# Wind Turbine Blade Cracking Detection under Imbalanced Data Using a Novel Roundtrip Auto-Encoder Approach

Yuyan Zhang [1] , Yafeng Zhang [1], Hao Li [1,*], Lingdi Yan [2], Xiaoyu Wen [1] and Haoqi Wang [1]

1   Henan Key Laboratory of Intelligent Manufacturing of Mechanical Equipment, Zhengzhou University of Light Industry, Zhengzhou 450003, China; 2020022@zzuli.edu.cn (Y.Z.); zhangyafeng1998@163.com (Y.Z.)
2   CITIC Heavy Industries Co., Ltd., Luoyang 471039, China
*   Correspondence: lh9666@hotmail.com

**Abstract:** Imbalanced data cause low recognition of wind turbine blade cracking. Existing data-level augmentation methods, including sampling and generative strategies, may yield lots of high-confidence but low-value samples, which fail to improve the detection of blade cracking. Therefore, this paper designs a novel RTAE (roundtrip auto-encoder) method. Based on the idea of the roundtrip approach, we design two generator networks and two discriminator networks to ensure the cycle mapping between cracking samples and latent variables. Further, by leveraging cycle consistency loss, generated samples fit the distribution of historical cracking samples well. Thus, these generated samples effectively realize data augmentation and improve recognition of blade cracking. Additionally, we apply an auto-encoder method to reduce the dimension of historical samples and thus the complexity of the generator network and discriminator network. Through the analysis of real wind turbine blade cracking data, the recognition of cracking samples is improved by 19.8%, 23.8% and 22.7% for precision, recall and F1-score.

**Keywords:** blade cracking; imbalanced data; roundtrip; auto-encoder

## 1. Introduction

Due to the insufficient bending strength, tensile strength and other parameters of wind turbine blade material, as well as their overly large dynamic load, blade cracking is a common cause of failure, accounting for approximately 30% of all downtime accidents [1–3]. However, due to constantly changing speeds and transient impacts, modeling the failure mechanism of blade cracking proves very challenging. Furthermore, since blades account for roughly 15–20% of total equipment costs, blade cracking can lead to potentially enormous maintenance expenses and safety hazards. Given its importance and costs, timely and accurate approaches to detecting blade cracking are attracting widespread research attention [4–6].

Data-based detection approaches work independently of the dynamics and kinematics of physical systems, and are thus effective for blade cracking detection. However, data are generally imbalanced, with very few cracking samples in real scenarios, leading to detection approaches that support normal samples and rarely identify cracking samples in the early stage (theoretical analysis in Section 2.1). Aiming at such problems, current data-level solutions mainly include sampling and generative strategies.

Sampling strategies, e.g., SMOTE (synthetic minority oversampling technique) and ADASYN (adaptive synthetic), utilize prior knowledge about the space distribution of collected cracking samples to synthesize new samples. Ge [7] adopted 29 sensor features as original input and used SMOTE to synthesize virtual cracking samples via linear interpolation between two random real cracking samples. Meanwhile, the density and concentration area are introduced to increase the confidence of virtual cracking samples. Jiang [8] designed a synthetic and dependent wild-bootstrapped oversampling technique for wind

turbine fault diagnosis which is a modification of SMOTE and mimics the temporal dependence of time-series data. In the data analysis part, two datasets collected from two wind farms of northeast and northwest China are used, with 70 attributes in total, including wind speed, x-axis nacelle vibration, yaw angle, etc. Cristian [9] dealt with imbalanced wind turbine data using the random oversampling technique, which directly removed some random normal samples to achieve amount balance. To verify the proposed method, 19 real variables covering 19 different attributes were used, including generated power, R-phase voltage, wind speed, etc. Yi [10] designed a minority clustering SMOTE approach for wind turbine fault detection, which achieved different clusters of minority samples and solved the problem of the uneven distribution of fault samples. Chen [11] adopted ADASYN to alleviate the critical imbalance and convolutional LSTM-GRU to recognize blade icing status. The dataset for verification was collected over 341.88 h with 26 sensor values, including wind speed, yaw position, vertical acceleration, etc.

Generative strategies estimate the probability density of cracking samples and then sample new data points to supplement cracking samples. Chen [12] designed a deep convolutional generative adversarial network (GAN) to produce a threshold for a condition monitoring scheme of wind turbines. The employed signal was the frequency spectrum transformed by fast Fourier transform. Liu [13] used a generative adversarial network to transform normal data into rough fault data, and furthermore, a refiner developed using a GAN was adopted to make them much similar to real fault data. For verification, the input data contained 28 variables, such as pitch angle, hub angle and generator torque. Liu [14] introduced sparse dictionary learning into an adversarial variational auto-encoder to generate virtual data and determine the posterior distribution from six sensor variables, including wind speed, active power, generator speed, three-phase current and voltage. Wang [15] designed a least-square GAN to determine the distribution of health data from 15 selected sensor variables and realize the data augmentation. Yang [16] used wavelet packet transform to generate time–frequency data of wind turbines and a GAN to compensate for the imbalance level. Ogaili [17] published a wind turbine fault diagnosis dataset considering vibration under different wind speeds for fault diagnosis under both imbalanced and balanced conditions. Jiang [18] took 28 sensor variables (e.g., yaw position, yaw speed) as input of a GAN to generate virtual blade icing samples. Zou [19] combined a convolution neural network and GAN to detect wind turbine blade surface damage using a small number of damage images.

Although plenty of approaches have been developed, there are still two critical problems that deserve in-depth research based on above studies. (1) Both sampling strategies and generative strategies easily produce virtual cracking samples with a high confidence but low usage value. The reason for this is that sampling strategies more often pick samples in a dense area as a basis, leading to many synthesized samples lying in the middle area. Similarly, the generator in a GAN yields virtual cracking samples in the middle area (with high confidence) to easily cheat the discriminator. However, virtual samples near the decision border rather than the middle area may be more helpful for supporting good classification results. (2) Generative strategies focus on generating real, similar cracking samples but ignore the overall space distribution of real samples. Furthermore, if real samples are distributed over more than one cluster, with some clusters owning many data points, generative strategies may only produce samples for these clusters. However, these clusters with few data points may be more important for improving classification accuracy.

Aiming at the above key points, this work designs a novel blade-cracking detection solution. Considering the case that the amount of redundant features will bring a high number of weights to be optimized in GAN, we curtail the input dimension using an unsupervised auto-encoder. Further, a virtual cracking sample generative strategy-based roundtrip framework is designed to achieve bidirectional mapping between virtual and real samples. By inverse mapping, it catches the overall space distribution and avoids the generated virtual samples falling into a dense area. Through verification on the benchmark dataset and real wind turbine blade cracking, the results show the effectiveness.

## 2. Motivation and Preliminary

### 2.1. Motivation

Wind turbine blade-cracking detection is typically a binary classification problem. In order to explain the effects of imbalanced data, this section takes two common binary classification methods for analysis, namely decision tree (C4.5) and support vector machine (SVM). For the former, a splitting node is selected according to the information gain ratio:

$$\text{Gain\_ratio} = \frac{\sum_{i=1}^{N} \frac{D_i}{D} \left( -\frac{D_i^+}{D_i} \log_2 \frac{D_i^+}{D_i} - \frac{D_i^-}{D_i} \log_2 \frac{D_i^-}{D_i} \right)}{\left( -\frac{D^+}{D} \log_2 \frac{D^+}{D} - \frac{D^-}{D} \log_2 \frac{D^+}{D} \right)}$$
$$\text{s.t. } D = D^+ + D^-, D_i = D_i^+ + D_i^-$$
$$D^+ \gg D^-, D_i^+ \gg D_i^- \tag{1}$$

where $D$ denotes the total sample amount; $D^+$ denotes the normal sample amount; $D^-$ denotes the blade cracking sample amount. $N$ is the total number of attribute values for a specific attribute. $D_i$ denotes the sample amount for $i$-th attribute value. $D_i^+$ and $D_i^-$ are the normal sample amount and cracking sample amount for $i$-th attribute value.

Since

$$\lim_{x \to 0^+} x \log_2(x) = \lim_{x \to 0^+} x \frac{\ln(x)}{\ln 2} = \frac{1}{\ln 2} \lim_{x \to 0^+} x \ln(x)$$
$$= \frac{1}{\ln 2} \lim_{x \to 0^+} \frac{\ln(x)}{1/x} = \frac{1}{\ln 2} \lim_{x \to 0^+} \frac{1/x}{-1/x^2} = -\frac{1}{\ln 2} \lim_{x \to 0^+} x = 0 \tag{2}$$

Supposing there are very few cracking samples, we can infer that

$$\text{Gain\_ratio} \approx \frac{\sum_{i=1}^{N} \frac{D_i}{D} \left( \frac{D_i^+}{D_i} \log_2 \left( \frac{D_i^+}{D_i} \right) \right)}{\left( \frac{D^+}{D} \log_2 \left( \frac{D^+}{D} \right) \right)} = \frac{\sum_{i=1}^{N} \frac{D_i^+}{D^+} \log_2 \left( \frac{D_i^+}{D_i} \right)}{\log_2 \left( \frac{D^+}{D} \right)} \tag{3}$$

From Equation (3), it is clear that gain ratio is irrelevant with cracking samples. That is, building the decision tree only depends on normal samples, causing a high recognition uncertainty for cracking samples.

For SVM, we take the linear separable case for explanation. If the data are balanced, the decision surface is obtained by minimizing Equation (4):

$$\underset{w}{argmin} \text{ Loss} = \| W \| + C \sum_{i=1}^{D} \xi_i, \text{ s.t. } y_i(Wx_i + b) \geq 1 - \xi_i, \xi_i \geq 0 \tag{4}$$

where $W$ decides the position and direction of decision surface; $C$ is a penalty coefficient; $\xi_i$ is a relaxation variable; $\{x_i, y_i\}$ denotes $i$-th sample. If some cracking samples are removed and surface shifts distance d towards cracking samples, the variation in *Loss* is computed by Equation (5):

$$\Delta Loss = C \sum_{i=1}^{D^+} (\xi_i - d\sin\theta) + C \sum_{j=1}^{D^-} (\xi_j + d\sin\theta) - \left( C \sum_{i=1}^{D^+} \xi_i + C \sum_{j=1}^{D^-} \xi_j \right)$$
$$= Cd\sin\theta(D^- - D^+) < 0 \tag{5}$$

where $\theta$ represents the angle between the x-axis and decision surface. Obviously, $\Delta Loss$ is less than zero, implying a better support for normal samples, but it may bring many misclassifications of these unseen cracking samples near the former decision surface.

From the above analysis, the root cause for low recognition of cracking samples is the imbalanced sample amount. Current solutions mainly include two aspects, namely sampling strategies and generative strategies. However, there are still critical problems that deserve deep research. As illustrated in Figure 1a, sampling strategies (e.g., SMOTE and its

variants) may pick the basis sample from a middle or dense area, leading to synthesized samples lying far from border. As shown in Figure 1b,c, generative strategies (such as GAN) tend to generate highly confident samples for easily cheating the discriminator. However, samples on the border with low confidence may be better for improving the decision surface. Additionally, in the case of several clusters (as Figure 1d), generative strategies may ignore cluster 1 because of less cracking samples, since it mainly fits the probability distribution of cluster 2. Instead, cluster 1 may be more important for improving classification accuracy.

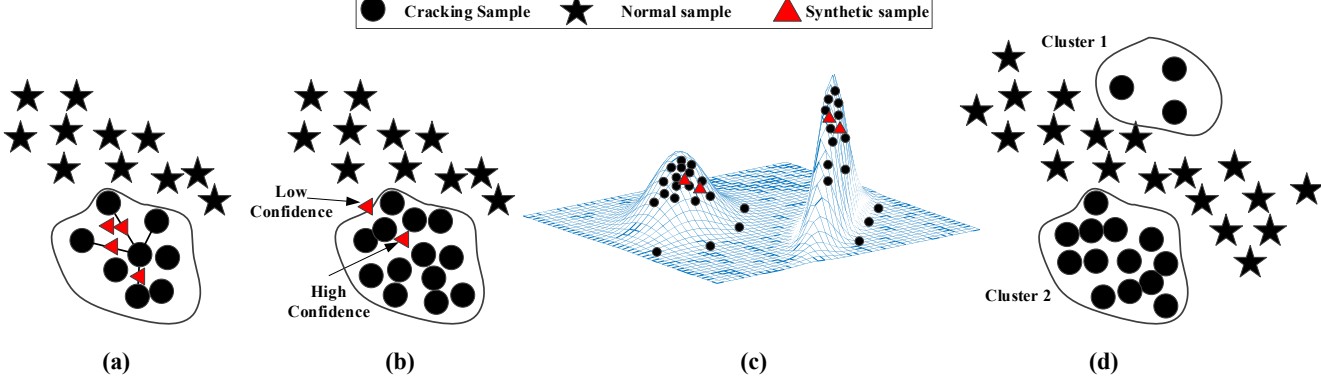

**Figure 1.** Illustration of problem about sampling strategy and generative strategy.

### 2.2. Preliminary

Roundtrip

Roundtrip is a method based on cycle GAN and its framework is shown in Figure 2 [20]. It contains four parts: generator network G and H, discriminators $D_X$ and $D_Z$. $Z$ is a latent variable drawn from normal distribution; $X$ denotes historical samples; $\widetilde{Z}$ and $\widetilde{X}$ denote generated latent variables and generated virtual samples. H transforms latent variables to virtual samples and G transforms historical samples to latent variables. Both of them are nonlinear transforms. $D_X$ is employed to discriminate real and virtual samples. Likewise, discriminator $D_Z$ discriminates real and virtual latent variables. $D_X$ and $D_Z$ determines their authenticity (real 0, fake 1). By minimizing the following six objective functions, G, H, $D_X$, and $D_Z$ achieve the optimal configuration.

$$\widetilde{Z} = H(X) \tag{6}$$

$$\widetilde{X} = G(Z) \tag{7}$$

$$
\begin{cases}
\min_{D_X} L_1 = E_{X \sim P(X)}\left[D_X^2(X)\right] + E_{Z \sim P(Z)}\left[(1 - D_X(\widetilde{X}))^2\right] \\
\min_{D_Z} L_2 = E_{X \sim P(X)}\left[(1 - D_Z(\widetilde{Z}))^2\right] + E_{Z \sim P(Z)}\left[D_Z^2(Z)\right] \\
\min_{G} L_3 = E_{Z \sim P(Z)}\left[D_X^2(\widetilde{X})\right] \\
\min_{H} L_4 = E_{X \sim P(X)}\left[D_z^2(\widetilde{Z})\right] \\
\min_{G,H} L_5 = E_{Z \sim P(Z)}\left[\| Z - H(G(Z)) \|_2^2\right] \\
\min_{G,H} L_6 = E_{X \sim P(X)}\left[\| X - G(H(X)) \|_2^2\right]
\end{cases}
\tag{8}
$$

where $L_1$ and $L_2$ aim to obtain well-performed discriminators; $L_3$ and $L_4$ supervise G and H to generate high-quality data, namely $\widetilde{Z}$ and $\widetilde{X}$; $L_5$ and $L_6$ measure the cycle consistency. It is noted that when $L_3$ and $L_4$ decrease, the generated data will move away from the

decision boundary and stay in high-density areas, which will produce data with high confidence but of low quality.

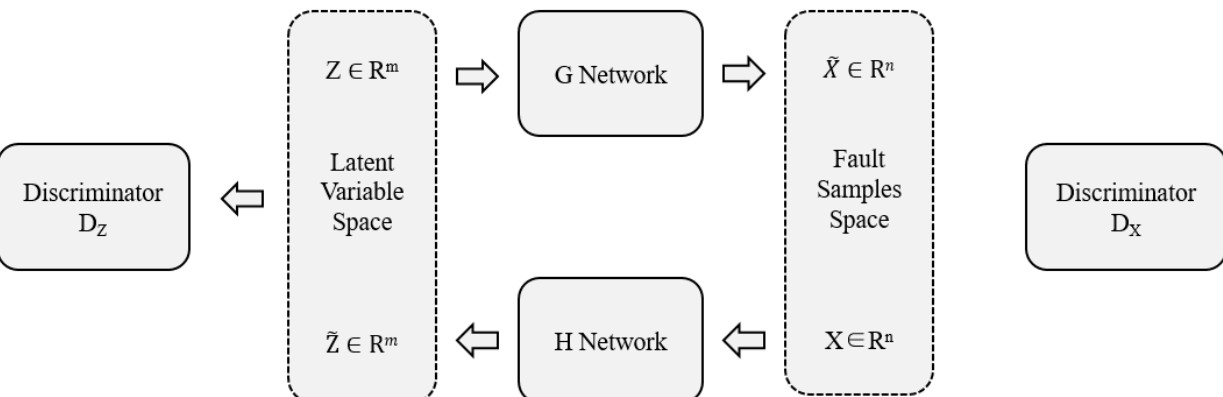

**Figure 2.** Overall framework of roundtrip.

## 3. RTAE Method

### 3.1. Intrinsic Feature Extracted by Auto-Encoder

The collected wind turbine blade cracking data are usually high dimensional (75-dimension in the case study). Directly inputting such data into roundtrip will bring big H and G networks, that is, a large number of hidden layers and hidden neurons. Generally, it is difficult to design such effective networks and configure them. Therefore, reducing the dimensionality before inputting into roundtrip is urgently necessary.

Auto-encoder (AE) is an outstanding dimension reduction method and effective for nonlinear problems [21]. It consists of an encoder and a decoder. The encoder encodes input into a feature representation and decoder reconstructs the input. Given the observations $Y \in \mathrm{R}^n$, the encoder computes low dimensional representation by

$$\widetilde{Y} = f(W_E Y + b_E) \tag{9}$$

where $f(\cdot)$ is activation function; $W_E \in \mathrm{R}^{m \times n}$ is an encoder matrix and $b_E \in \mathrm{R}$ is a bias; $n$ is the dimension of original sample. Decoder reconstructs it by:

$$Y = f(W_D \widetilde{Y} + b_D) \tag{10}$$

where $W_D \in \mathrm{R}^{n \times m}$ is a decoder matrix; $b_D \in \mathrm{R}$ is a bias; $m$ is the low dimension. AE learns these matrixes and biases by

$$arg \min_{\boldsymbol{\theta}} \frac{1}{2K} \sum_{i=1}^{K} \parallel \widetilde{Y}_i - Y_i \parallel_2^2, \boldsymbol{\theta} = \{W_E, b_E, W_D, b_D\}, i = 1, 2, \dots, K \tag{11}$$

where $K$ denotes the sample amount.

### 3.2. Implementation of Roundtrip Density Estimation

For roundtrip, H, G, $\mathrm{D_X}$ and $\mathrm{D_Z}$ are the key components. Based on the dataset used in the data analysis part (each sample contains 75 attributes), we designed these four models as listed in Table 1. As shown, all of these four models contain five layers, with an alternating fully connected layer and batch normalization.

**Table 1.** Details of H, G, D$_X$ and D$_Z$.

| Part | Layer Type | Input Dimension (*IP*) | Output Dimension (*OP*) |
|---|---|---|---|
| H | Fully Connected Layer | Determined by AE | 32 |
| | Batch Normalization | 32 | 32 |
| | Fully Connected Layer | 32 | 64 |
| | Batch Normalization | 64 | 64 |
| | Fully Connected Layer | 64 | Dimension of Latent Variable |
| G | Fully Connected Layer | Dimension of Latent Variable | 64 |
| | Batch Normalization | 64 | 64 |
| | Fully Connected Layer | 64 | 32 |
| | Batch Normalization | 32 | 32 |
| | Fully Connected Layer | 32 | Determined by AE |
| D$_X$ | Fully Connected Layer | Determined by AE | 32 |
| | Batch Normalization | 32 | 32 |
| | Fully Connected Layer | 32 | 16 |
| | Batch Normalization | 16 | 16 |
| | Fully Connected Layer | 16 | 2 |
| D$_Z$ | Fully Connected Layer | Dimension of Latent Variable | 20 |
| | Batch Normalization | 20 | 20 |
| | Fully Connected Layer | 20 | 10 |
| | Batch Normalization | 10 | 10 |
| | Fully Connected Layer | 10 | 2 |

For the fully connected layer, the computation formula is as follows:

$$X_{out} = WX_{in} + b \tag{12}$$

where $X_{in} \in R^{IP}$ denotes input for the current layer; $X_{out} \in R^{OP}$ denotes output of the current layer; $W \in R^{OP \times IP}$ represents a transformation matrix; $b$ is a bias; $IP$ and $OP$ are shown in Table 1.

For batch normalization, the computation formula is:

$$
\begin{aligned}
\mu &\leftarrow \frac{1}{m} \sum_{i=1}^{K} X_{ij} \\
\sigma^2 &\leftarrow \frac{1}{K} \sum_{i=1}^{K} \left( X_{ij} - \mu \right)^2 \\
\widetilde{X}_{ij} &\leftarrow \frac{X_{ij} - \mu}{\sqrt{\sigma^2 + \epsilon}} \\
\hat{X}_{ij} &\leftarrow \gamma \widetilde{X}_{ij} + \beta
\end{aligned}
\tag{13}
$$

where $X$ denotes the input batch; $X_{ij}$ denotes the $j$th dimension of $i$th sample; $\mu$ denotes the average value of $j$th dimension over all the samples in a batch; $\sigma^2$ denotes the variance for the $j$th dimension; $\epsilon$ is a small value avoiding denominator to be zero; $\gamma$ and $\beta$ are scale and translation factors. $\hat{X}_{ij}$ is the output of batch normalization for the $j$th dimension.

### 3.3. Alternating Gradient Descent-Based Training Process

To train H, G, D$_X$ and D$_Z$, we adopt an alternating gradient descent method. First, for D$_X$ and D$_Z$, we train them by the following formula:

$$\boldsymbol{\theta}_{D_X} = \boldsymbol{\theta}_{D_X} - lr \frac{\partial L_1}{\partial \boldsymbol{\theta}_{D_X}} \tag{14}$$

$$\boldsymbol{\theta}_{D_Z} = \boldsymbol{\theta}_{D_Z} - lr \frac{\partial L_2}{\partial \boldsymbol{\theta}_{D_Z}} \tag{15}$$

where $\boldsymbol{\theta}_{D_X}$ and $\boldsymbol{\theta}_{D_Z}$ are, respectively, the parameters from the network in Table 1. Specifically, they are the weights and bias from (12); $lr$ is the learning rate; $L_1$ and $L_2$ are the loss function in Equation (8).

Second, to optimize G and H synchronously, we rewrite the loss function in Equation (8) as:

$$\min_{G,H} L = L_3 + L_4 + \alpha(L_5 + L_6) \tag{16}$$

where $L$ is the overall loss function; $\alpha$ is a pre-specified coefficient. Finally, the weights and bias in G and H are updated by the following equation:

$$\boldsymbol{\theta}_{G,H} = \boldsymbol{\theta}_{G,H} - lr\frac{\partial L}{\partial \boldsymbol{\theta}_{G,H}} \tag{17}$$

### 3.4. Overall Framework

The overall framework of the proposed RTAE method is shown in Figure 3. The specific steps are:

(1)　Obtain normal and cracking samples;
(2)　Input both cracking and normal samples into auto-encoder for dimension reduction;
(3)　Input low-dimensional representation cracking samples into roundtrip for data augmentation;
(4)　Generate cracking samples and fuse them with these original samples (after dimension reduction);
(5)　Implement classification result analysis.

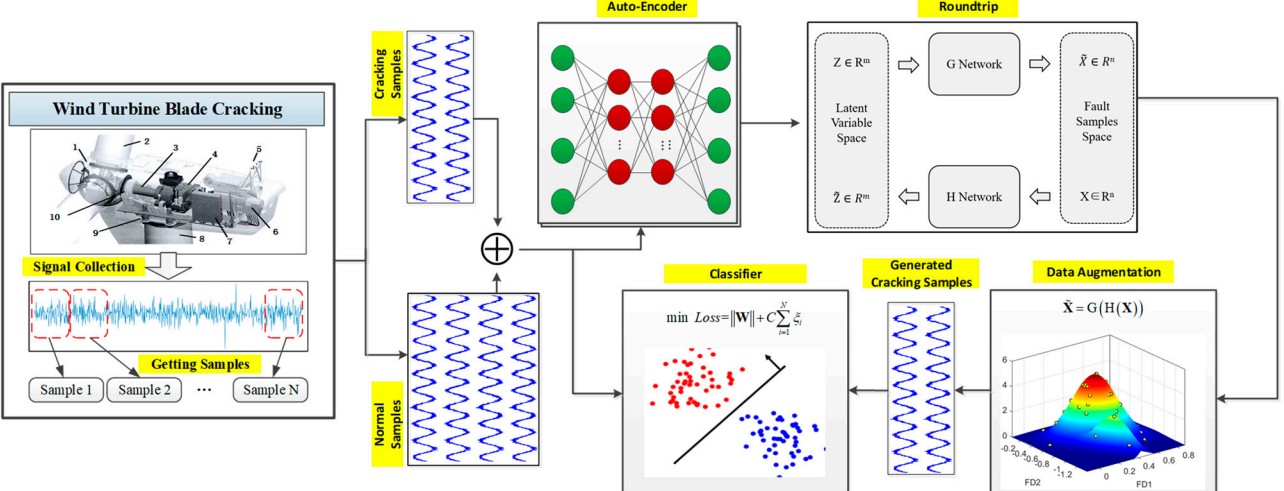

**Figure 3.** Overall framework of RTAE.

The detailed steps are listed in Table 2. It is noted that we can compare the results 'before data augmentation' and 'after data augmentation' to verify the effectiveness of RTAE.

**Table 2.** Detailed steps of RTAE method.

| **Input:** | **Original Samples $Y=\{Y_1,\ldots,Y_K\}$ Contain Both Normal and Cracking Samples** |
|---|---|
| Step1: | Minimize $\underset{\theta}{argmin}\ \frac{1}{2K}\sum_{i=1}^{K}\parallel \widetilde{Y}_i - Y_i \parallel_2^2$ to obtain $\{W_E, b_E\}$ |
| Step2: | According to $\widetilde{Y} = f(W_E Y + b_E)$, compute low-dimensional representation. |
| Step3: | For cracking and normal samples, we name them as $\widetilde{Y}_{min}$ and $\widetilde{Y}_{maj}$, respectively. |
| Step4: | Replace $X$ in Figure 2 by $\widetilde{Y}_{min}$. $Z$ is set as the same dimension as $X$. Each dimension of $Z$ is a random variable drawn from normal distribution with mean value 0.5, standard deviation 0.05. |

**Table 2.** *Cont.*

| Input: | Original Samples $Y=\{Y_1,\dots,Y_K\}$ Contain Both Normal and Cracking Samples |
| --- | --- |
| Step5: | Train H, G, $D_X$ and $D_Z$ by (14), (15) and (17). |
| Step6: | Compute $\widetilde{X} = G(H(X))$. |
| Step7: | Implement 5-fold cross-validation and compute the average validation result. |
| Step8: | Take the result only on $\left\{\widetilde{Y}_{min}, \widetilde{Y}_{maj}\right\}$ as 'before data augmentation'. For each fold train process, adding $\widetilde{X}$ to the train set and take the result as 'after data augmentation'. |

## 4. Engineering Applications

### 4.1. Description of Blading Cracking Dataset

The datasets used for analysis were collected from the CCF Big Data & Computing Intelligence Contest (https://www.datafountain.cn/competitions/302/datasets (accessed on 28 June 2018)). These data were collected by supervisory control and data acquisition system under running status. Samples are collected every 10 min, with each sample containing 75 attributes, such as blade angle, over-speed sensor value, x-axis vibration, inverter power voltage, etc. Details can be found in Table 3. There are in total 48,340 samples collected during approximately one year with a majority of normal samples. To verify the effectiveness under a different imbalance ratio, we constructed 10 datasets as listed in Table 4. We randomly selected 2000 normal samples and varying number of cracking samples. The sign '#' in Table 4 denotes the sample amount. The imbalance ratio varies from 20:1 to 2:1.

**Table 3.** Detailed attributes of the used data.

| No. | Attribute | No. | Attribute | No. | Attribute |
| --- | --- | --- | --- | --- | --- |
| 1 | Hub speed | 26 | Inverter inlet temperature | 51 | Variable motor 1 Power estimation |
| 2 | Hub angle | 27 | Converter outlet temperature | 52 | Variable motor 2 Power estimation |
| 3 | Blade 1 angle | 28 | Inverter inlet pressure | 53 | Variable motor 3 Power estimation |
| 4 | Blade 2 angle | 29 | Converter outlet pressure | 54 | Current status of the blade |
| 5 | Blade 3 angle | 30 | Generator power limiting value | 55 | Current hub status value |
| 6 | Variable pitch motor1 Current | 31 | Reactive power set value | 56 | Yaw status value |
| 7 | Variable pitch motor2 Current | 32 | Rated hub speed | 57 | Yaw requirement value |
| 8 | Variable pitch motor3 Current | 33 | Ambient temperature of the wind tower | 58 | Blade 1 Battery box temperature |
| 9 | Detection value of the overspeed sensor | 34 | Generator stator temperature 1 | 59 | Blade 2 Battery box temperature |
| 10 | Average of 5 s yaw against wind | 35 | Generator stator temperature 2 | 60 | Blade 3 Battery box temperature |
| 11 | Vibration value in x direction | 36 | Generator stator temperature 3 | 61 | Blade 1 Variable motor temperature |
| 12 | Vibration value in y direction | 37 | Generator stator temperature 4 | 62 | Blade 2 Variable motor temperature |
| 13 | Hydraulic brake pressure | 38 | Generator stator temperature 5 | 63 | Blade 3 Variable motor temperature |
| 14 | Cabin weather station wind speed | 39 | Generator stator temperature 6 | 64 | Blade 1 Inverter box temperature |
| 15 | Absolute wind direction | 40 | Generator air temperature 1 | 65 | Blade 2 Inverter box temperature |
| 16 | Atmospheric pressure | 41 | Generator air temperature 2 | 66 | Blade 3 Inverter box temperature |
| 17 | Reactive power control status | 42 | Main bearing temperature 1 | 67 | Blade 1 Super-capacitor voltage |
| 18 | Inverter power grid side current | 43 | Main bearing temperature 2 | 68 | Blade 2 Super-capacitor voltage |

**Table 3.** *Cont.*

| No. | Attribute | No. | Attribute | No. | Attribute |
|---|---|---|---|---|---|
| 19 | Inverter power grid side voltage | 44 | Hub temperature | 69 | Blade 3 Super-capacitor voltage |
| 20 | Active power of inverter power grid side | 45 | Hub control cabinet temperature | 70 | Drive 1 Thyristor temperature |
| 21 | Inverter grid side reactive power | 46 | Cabin temperature | 71 | Drive 2 Thyristor temperature |
| 22 | Inverter generator side power | 47 | Control cabinet temperature | 72 | Drive 3 Thyristor temperature |
| 23 | Generator operating frequency | 48 | Inverter INU temperature | 73 | Drive 1 Output torque |
| 24 | Generator current | 49 | Inverter ISU temperature | 74 | Drive 2 Output torque |
| 25 | Generator torque | 50 | Inverter INU RMIO temperature | 75 | Drive 3 Output torque |

**Table 4.** Details of 10 datasets.

| No. | Dataset | # Normal | # Cracking | Imbalance Ratio |
|---|---|---|---|---|
| 1 | Wind_5 | 2000 | 100 | 20:1 |
| 2 | Wind_10 | 2000 | 200 | 10:1 |
| 3 | Wind_15 | 2000 | 300 | 6.67:1 |
| 4 | Wind_20 | 2000 | 400 | 5:1 |
| 5 | Wind_25 | 2000 | 500 | 4:1 |
| 6 | Wind_30 | 2000 | 600 | 3.33:1 |
| 7 | Wind_35 | 2000 | 700 | 2.86:1 |
| 8 | Wind_40 | 2000 | 800 | 2.5:1 |
| 9 | Wind_45 | 2000 | 900 | 2.22:1 |
| 10 | Wind_50 | 2000 | 1000 | 2:1 |

Figure 4 shows two normal samples as well as two cracking samples. For each status, these two samples are collected from two different time points. Obviously, it is difficult to distinguish blade status with only one or just a few attributes, since samples from different time points present different value distributions. From such results, we can infer that samples from the same class distribute over more than one cluster. Applying SMOTE or GAN to enhance the cracking sample amount may fail.

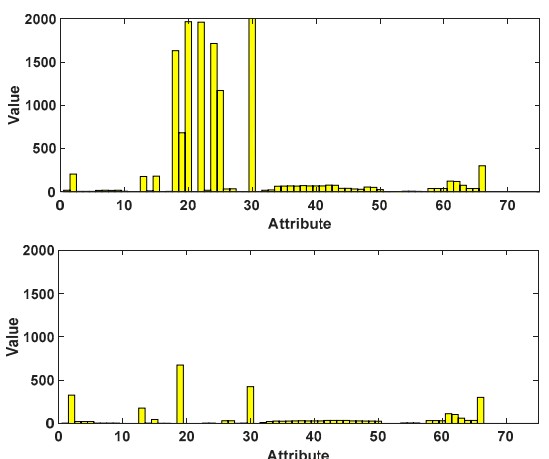
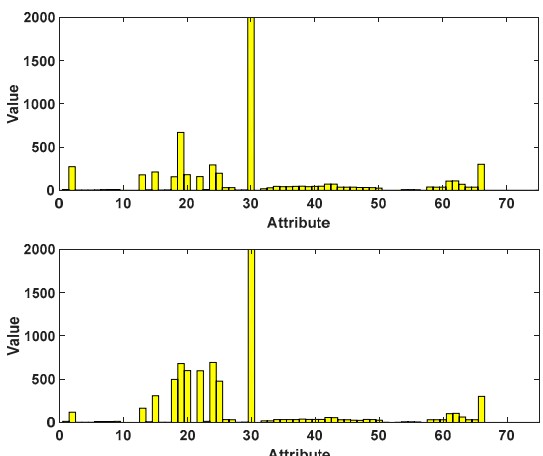

**Figure 4.** Normal sample (**first row**) and cracking sample (**second row**).

### 4.2. Sample Balanced by RTAE

Due to different units, the attribute ranges vary significantly. Normalization is used to scale the range of each attribute to [0, 1]. The normalization method is presented below:

$$\widetilde{A} = \frac{A - A_{min}}{A_{max} - A_{min}} \tag{18}$$

where $\tilde{A}$ is the normalized attribute value; $A_{min}$ and $A_{max}$ are, respectively, the minimum and maximum values among all the samples.

Auto-encoder is adopted to extract intrinsic features and reduce the dimension. Wind_50 dataset is used to train an AE model, with its structure [75, 20, 10]. The encoder include two hidden layers, with twenty units for the first layer and ten units for the second layer. Sigmoid is taken as the activation function. For the decoder, it has the inverse structure, namely, as shown in [10, 20, 75]. Figure 5 shows four samples and their reconstructed results. Obviously, the reconstructed data approximate original samples, implying the intrinsic feature can adequately represent the key information.

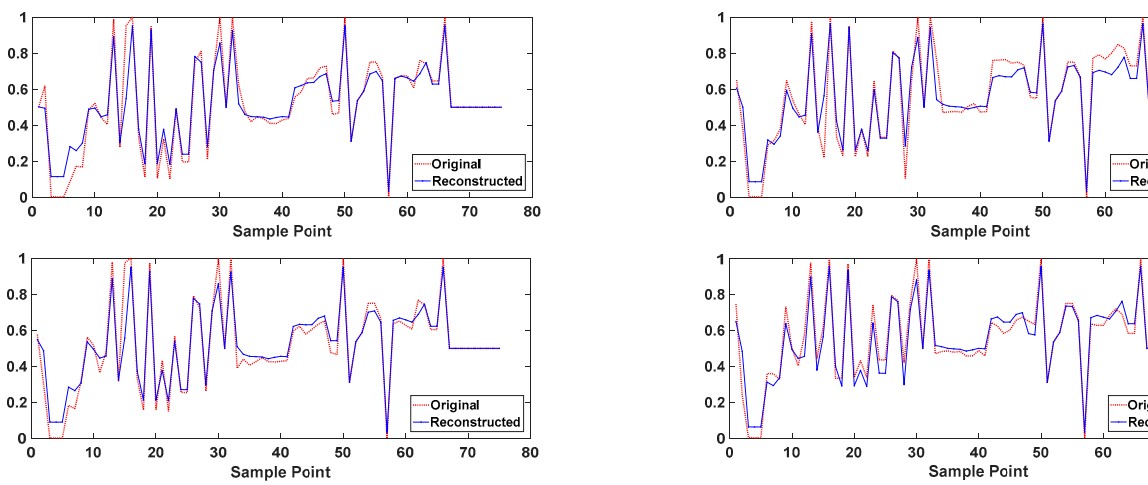

**Figure 5.** Four original samples and their reconstructed result.

Roundtrip is employed to generate virtual cracking samples. Table 5 lists detailed hyper-parameter settings. These parameters are determined by trial and error. To observe the results of roundtrip, loss values during 5000 iterations are shown in Figure 6. $L_1$ and $L_2$ in Equation (8) are relevant to the generator, thus they are added together. Both the losses of the generator and discriminator go down very fast and converge after 2000 iterations. In the last 100 iterations (see the inset window), both of them fluctuate to a small extent. When the red line goes down, blue line will go up, implying adversarial learning. To observe generated data, TSNE [22] is adopted to map the intrinsic features into two-dimension space. The results are shown in Figure 7. The generated data well fit the original data distribution and many samples lie on the border, which is helpful for improving classification accuracy.

**Table 5.** Details of hyper-parameter values.

| Hyper Parameter | Symbol | Value |
|---|---|---|
| Factor in (13) | $\varepsilon$ | 0.001 |
| Scale factor in (13) | $\gamma$ | 1 |
| Translation factor in (13) | $\beta$ | 0 |
| Learning rate in (14), (15) and (17) | $lr$ | 0.0002 |
| Weight in (16) | $\alpha$ | 10 |
| Iteration Number | - | 5000 |

### 4.3. Effectiveness of Data Augmentation

Table 6 lists the comparisons between, with and without data augmentation. Decision tree is adopted to classify normal and cracking samples [23]. Three metrics are adopted to evaluate the performance, namely precision, recall and F1-score. Detailed metric computations can be found in the literature [24]. For normal samples, the three metrics with data augmentation are slightly lower than or equal to those without data augmentation. However, for cracking samples, these metrics are higher. For instance, 'Wind_35' achieves

precision (0.73), recall (0.76) and F1-score (0.75) with data augmentation, which are slightly lower than that without data augmentation (precision: 0.74, recall: 0.78, F1-score: 0.76). The recall value improves from 0.22 to 0.26 and F1-score increases from 0.23 to 0.25, implying the effectiveness of data augmentation. With the data augmentation approach, the average value of the cracking samples are 0.133, 0.135 and 0.135, meaning 19.8%, 23.8% and 22.7% improvement on these three metrics.

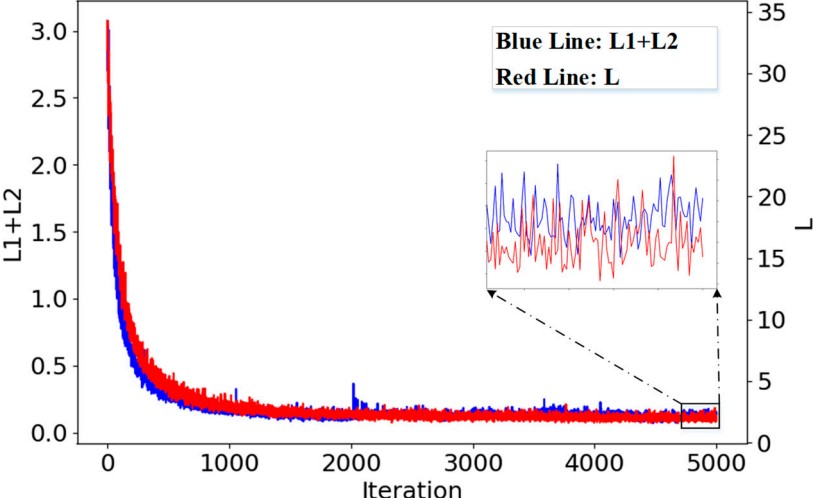

**Figure 6.** Loss value change during the iteration.

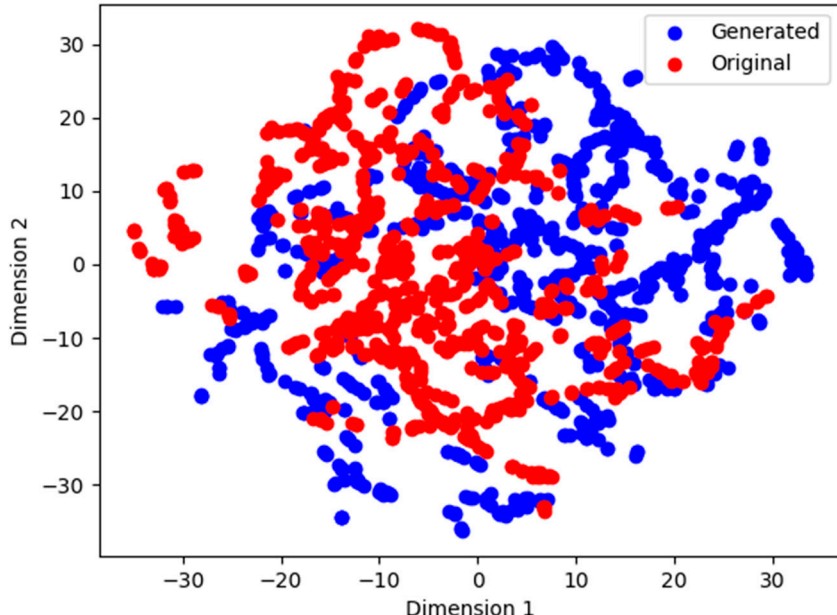

**Figure 7.** Four original samples and their reconstructed results.

We draw the results with data augmentation in histograms, as shown in Figure 8. For normal samples, these three metrics draw down. This is because too few cracking samples will be overwhelmed and nearly all these samples are classified into normal samples. Therefore, from Wind_5 to Wind_35, the recall value is bigger than precision (note that the F1-score is determined by the former two metrics). With an increase in the amount of cracking samples, some normal samples are misclassified into cracking samples, causing a small recall value. For cracking samples, with the imbalance ratio alleviating, all the three metrics increase. It can be concluded that RTAE is effective for improving the recognition of wind blade cracking samples.

**Table 6.** Comparison between, with and without data augmentation.

| Dataset | | Without Data Augmentation | | | With Data Augmentation | | |
|---|---|---|---|---|---|---|---|
| | | Precision | Recall | F1-Score | Precision | Recall | F1-Score |
| Wind_5 | Normal | 0.95 | 0.97 | 0.96 | 0.95 | 0.96 | 0.96 |
| | Cracking | 0 | 0 | 0 | 0.02 | 0.02 | 0.02 |
| Wind_10 | Normal | 0.90 | 0.93 | 0.91 | 0.90 | 0.91 | 0.91 |
| | Cracking | 0 | 0 | 0 | 0.02 | 0.02 | 0.02 |
| Wind_15 | Normal | 0.86 | 0.89 | 0.87 | 0.85 | 0.85 | 0.85 |
| | Cracking | 0 | 0 | 0 | 0.02 | 0.02 | 0.02 |
| Wind_20 | Normal | 0.81 | 0.85 | 0.83 | 0.81 | 0.85 | 0.83 |
| | Cracking | 0 | 0 | 0 | 0.04 | 0.03 | 0.04 |
| Wind_25 | Normal | 0.77 | 0.83 | 0.80 | 0.77 | 0.82 | 0.79 |
| | Cracking | 0 | 0 | 0 | 0.05 | 0.04 | 0.05 |
| Wind_30 | Normal | 0.79 | 0.81 | 0.78 | 0.76 | 0.77 | 0.76 |
| | Cracking | 0.21 | 0.17 | 0.19 | 0.18 | 0.17 | 0.18 |
| Wind_35 | Normal | 0.74 | 0.78 | 0.76 | 0.73 | 0.76 | 0.75 |
| | Cracking | 0.25 | 0.22 | 0.23 | 0.24 | 0.26 | 0.25 |
| Wind_40 | Normal | 0.69 | 0.69 | 0.69 | 0.71 | 0.71 | 0.71 |
| | Cracking | 0.21 | 0.21 | 0.21 | 0.27 | 0.26 | 0.26 |
| Wind_45 | Normal | 0.66 | 0.62 | 0.64 | 0.66 | 0.64 | 0.65 |
| | Cracking | 0.25 | 0.29 | 0.27 | 0.26 | 0.28 | 0.27 |
| Wind_50 | Normal | 0.59 | 0.58 | 0.58 | 0.61 | 0.59 | 0.60 |
| | Cracking | 0.19 | 0.20 | 0.20 | 0.23 | 0.25 | 0.24 |
| Average | Cracking | 0.111 | 0.109 | 0.11 | 0.133 | 0.135 | 0.135 |

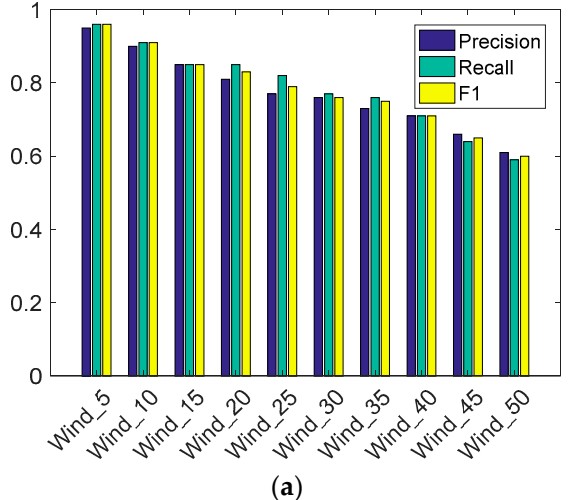

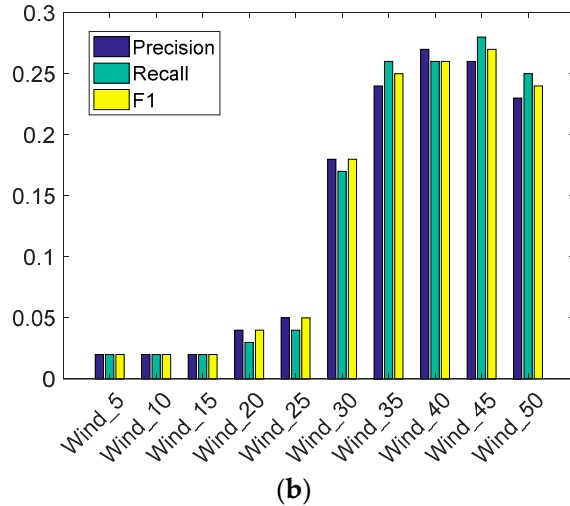

**Figure 8.** Result with data augmentation. (**a**) Normal Sample. (**b**) Cracking Sample.

### 4.4. Comparison with Other Imbalance Learning Methods

TRAE is compared to other two representative methods, GAN and SMOTE. Likewise, three metrics are used to evaluate the performance. For SMOTE, the number of the neighbor is set as 3, which is a commonly used value. For GAN, generator network and discriminator network are set as the same structure as in the literature [25]. The comparisons are listed in Table 7. RTAE outperforms GAN and SMOTE under all the 10 datasets. In particular, for Wind_35, Wind_40 and Wind_45, RTAE is far better than GAN and SMOTE. The main reason is that GAN focuses on 'real' or 'fake', which generates samples with high confidence, but low usage for improving the recognition of cracking samples. SMOTE cannot handle multi clusters. Therefore, we use a small neighbor number (i.e., 3) to alleviate such a situation. However, such a small value may make it pick the basis sample from the middle or a dense area, leading to many synthesized samples lying far from border.

**Table 7.** Comparison with GAN and SOMTE.

| Dataset | | RTAE Precision | Recall | F1 | GAN Precision | Recall | F1 | SMOTE Precision | Recall | F1 |
|---|---|---|---|---|---|---|---|---|---|---|
| Wind_5 | Cracking | 0.02 | 0.02 | 0.02 | 0 | 0 | 0 | 0 | 0 | 0 |
| Wind_10 | Cracking | 0.02 | 0.02 | 0.02 | 0 | 0 | 0 | 0 | 0 | 0 |
| Wind_15 | Cracking | 0.02 | 0.02 | 0.02 | 0 | 0 | 0 | 0 | 0 | 0 |
| Wind_20 | Cracking | 0.04 | 0.03 | 0.04 | 0 | 0 | 0 | 0 | 0 | 0 |
| Wind_25 | Cracking | 0.05 | 0.04 | 0.05 | 0 | 0 | 0 | 0 | 0 | 0 |
| Wind_30 | Cracking | 0.18 | 0.17 | 0.18 | 0.16 | 0.17 | 0.16 | 0.15 | 0.17 | 0.16 |
| Wind_35 | Cracking | 0.24 | 0.26 | 0.25 | 0.22 | 0.22 | 0.22 | 0.22 | 0.22 | 0.22 |
| Wind_40 | Cracking | 0.27 | 0.26 | 0.26 | 0.21 | 0.22 | 0.21 | 0.21 | 0.21 | 0.21 |
| Wind_45 | Cracking | 0.26 | 0.28 | 0.27 | 0.24 | 0.27 | 0.25 | 0.23 | 0.25 | 0.24 |
| Wind_50 | Cracking | 0.23 | 0.25 | 0.24 | 0.18 | 0.19 | 0.18 | 0.19 | 0.19 | 0.19 |

## 5. Conclusions and Future Work

Aiming at wind turbine blade cracking recognition under imbalanced data, this paper designs a novel roundtrip auto-encoder method. Two generator networks and two discriminator networks are designed to ensure the generated samples well fit the distribution of historical cracking samples. Auto-encoder method is applied to reduce the dimension of historical samples and thus the complexity of generator and discriminator. From the reconstructed results, it is concluded that auto-encoder is effective for extracting low-dimensional intrinsic features. From the results under different imbalance levels, the detection performance shows significant improvement with the RTAE method. When the imbalance level is above 4:1, all the cracking samples cannot be identified without data augmentation, which shows the importance of considering the imbalance problem in real application.

Analysis of the real wind turbine blade cracking data is carried out, the recognition of cracking samples improves by 19.8%, 23.8% and 22.7% on precision, recall and F1-score. The cracking detection under imbalanced data and the comparisons show that: (1) compared to the popular data-level methods, RTAE is superior under the influence of highly imbalanced data; (2) through integrating auto-encoder and roundtrip model, the framework provides a possibility for solving imbalance problem with high dimension data.

As is known, a wind turbine blade contains many types of cracks in real industrial applications. Currently, subject to roundtrip, RTAE can only deal with the binary-class imbalanced data problems. Multi-class cracking problems are more complex as they may contain more than one minority classes. In the future work, we will extend the application of RTAE and apply it to multi-class problem. Meanwhile, we will explore more effective roundtrip structures and apply them to high-dimensional time-series data.

**Author Contributions:** Conceptualization, Y.Z. (Yuyan Zhang); methodology, Y.Z. (Yafeng Zhang); software, L.Y.; vali-dation, Y.Z. (Yuyan Zhang) and Y.Z. (Yafeng Zhang); formal analysis, Y.Z. (Yuyan Zhang) and H.W.; investigation, Y.Z. (Yuyan Zhang) and H.W.; resources, H.L.; data curation, H.L.; writ-ing—original draft preparation, Y.Z. (Yuyan Zhang); writing—review and editing, Y.Z. (Yafeng Zhang); visualization, X.W.; supervision, H.L.; project administration, X.W.; funding acquisition, Y.Z. (Yuyan Zhang). All authors have read and agreed to the published version of the manuscript.

**Funding:** This research was funded by the National Natural Science Foundation of China (52105536); the Guangdong Basic and Applied Basic Research Fund (2022A1515140066); the Henan Provincial Key R&D and Promotion Special Project (Science and Technology Research) (232102221009).

**Data Availability Statement:** Data are contained within the article. The data presented in this study are available in Section 4.1.

**Conflicts of Interest:** Author Lingdi Yan was employed by the company CITIC Heavy Industries Co., Ltd. The remaining authors declare that the research was conducted in the absence of any commercial or financial relationships that could be construed as a potential conflict of interest.

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
