# Peer review of "Wind Turbine Blade Cracking Detection under Imbalanced Data Using a Novel Roundtrip Auto-Encoder Approach"

_applsci, doi:10.3390/app132111628_

Round 1
Reviewer 1 Report
The paper suggests improvement of crack detection in wind turbine blades and practically did not present review of methods of crack detection in blades.
The presented results are based on data collected from CCF Big Data & Computing Intelligence Contes ( https ://www.datafountain.cn/competitions/302/datasets) that are not available in English.
I would suggest the following paper improvements:
11. Provide the review of ML methods for crack detection in turbine blades with detail description of sensors, collected data and attributes.
22. Give information and the presented Chinese data base. Explain what data are included and how were they collected. Include information of the similar data bases that are acceptable in English.
33. Explain all applied attributes for the presented examples of developed method. I do not think that this data analysis can be called “experiment” because no experiments were conducted and the collected data were used for analysis.
Reviewer 2 Report
The manuscript, which I reviewed, proposes a method called Roundtrip Auto Encoder (RTAE) on the data sets of the wind turbine blade cracking. The authors claimed improvement in the recognition of the cracking samples. Following are my comments and suggestions for improvement:
1. Firstly, which types of cracks are recognized by this method? As the cracking in the wind turbine blade is multiscale because they are made of composite materials. As the proposed method is not based on the composite mechanics but still, I suggest there shall be literature discussing the cracking in the materials from which wind turbine blades are manufactured. Authors may read and cite the following articles for the cracking in composites:
https://doi.org/10.1016/j.compstruct.2018.06.010
https://doi.org/10.1016/j.triboint.2019.03.010
2. I suggest the authors shall use the format of the MDPI journals for the heading and subheadings rather than preliminary, etc.
3. The conclusions shall be extended as these seems insufficient.
4. What is the limitation of the proposed method?
5. State the novelty by comparing to the state-of-art methods.
Moderate level of Langauge is needed.
Round 2
Reviewer 1 Report
The paper improvements are useful and they brought more questions. More information about the windmill dataset is required.
The attributes presented in the Tabl.3 are needed to be explained. Where the presented attributes were measured, what are ranges of the measured parameters?
There are few duplications ## 14-15, 32-33 in the table.
Many of the same attributes have different numbers that are not explained. There are six generator stator temperatures (attributes 35-40), two main bearing temperatures, two generator air temperatures, three variable motor power estimations.
The presented table does not tell how many wind turbines are included to the dataset used for analysis and how many of them were damaged? What kind of damage did they have?
Does this set include data for turbines that were damaged during exploitation? What are differences in attributes before and after damage?
Reviewer 2 Report
I have no more comments!
Thanks!
Minor Grammar and Spelling check is needed.
Author Response
We have checked the grammar and spelling. Thanks for your comments.